# Spatio-Temporal Heterogeneity of Carbon Emissions and Its Key Influencing Factors in the Yellow River Economic Belt of China from 2006 to 2019

**DOI:** 10.3390/ijerph19074185

**Published:** 2022-03-31

**Authors:** Jingxue Zhang, Yanchao Feng, Ziyi Zhu

**Affiliations:** Business School, Zhengzhou University, Zhengzhou 450001, China; 202011011010004@gs.zzu.edu.cn (J.Z.); zhuziyi21@stu.zzu.edu.cn (Z.Z.)

**Keywords:** spatio-temporal heterogeneity, carbon emissions, Yellow River Economic Belt, spatial spillover effect

## Abstract

The Yellow River Economic Belt (YREB) performs an essential function in the low-carbon development of China as an important ecological protection barrier, and it is of great importance to identify its spatio-temporal heterogeneity and key influencing factors. In this study, we propose a comprehensively empirical framework to conduct this issue. The STIRPAT model was applied to determine the influencing factors of carbon emissions in the YREB from 2006 to 2019. The results show that the carbon emissions in the YREB had significant clustering characteristics in the spatial auto-correlation analysis. In addition, the estimation results of the spatial panel analysis demonstrate that the carbon emissions showed a distinct spatial lag effect and temporal lag effect. Moreover, the three traditional factors including population, affluence, technology are identified as the key influencing factors of carbon emissions in the YREB of China. Furthermore, the spatio-temporal heterogeneity is illustrated vividly by employing the GTWR-STIRPAT model. Finally, policy implications are provided to respond to the demand for low-carbon development.

## 1. Introduction

With the rapid growth of carbon dioxide emission in the process of urbanization, greenhouse gases are skyrocketing and triggering knock-on effects across ecosystems [1]. For instance, the melting sea caused by carbon sinks diminishing and climate warming has reduced the amount of land and threatened the survival of organisms [2]. Against this background, climate change has become a global dilemma, and the carbon emission of China becomes a hot spot for academia since it is the greatest emitter in the world [3]. To comply with the global trend of low-carbon development, the Chinese government announced that China will reach its carbon peak before 2030, which not only contributes to highlighting China’s reputation of shouldering international responsibility but also benefits China’s sustainable development strategy [4,5]. In particular, the Chinese government has implemented several well-designed measures, such as establishing seven regional pilot emissions trading schemes (ETS) since 2013 and launching the online trading of the national carbon market in July 2021, which can not only mitigate the cost of carbon change but also promote the efficiency of carbon productivity [6].

Carbon emissions in China have emerged as a hot spot for academia, in recent years, a number of studies on the influencing factors of it have been published [7,8,9]. Nevertheless, the spatio-temporal heterogeneity of carbon emissions was often ignored in the empirical analysis, which to some extent restricted the practical value of those studies. Hence, identifying the spatio-temporal heterogeneity and key influencing factors of carbon emissions simultaneously is critical and necessary for realizing China’s emission reduction and energy saving strategy [10]. With the development of the industrial chain in the new era, urban agglomeration has become the major carrier of carbon emissions in space [11,12]. As for the scale of the Chinese mainland, most studies focus on exploring the carbon emissions in the Beijing-Tianjin-Hebei [13], the Yangtze River Delta [14], and the Pearl River Delta [15,16], while few have paid attention to the Yellow River Economic Belt (YREB), which used to be the political and cultural center and related to the long-term revival of China [17]. However, compared with the BTH, YRD, and PRD, the economic development level of the YREB lags behind, with its industrial structure dominated by secondary industries [18]. Therefore, against the background of ecological civilization, the YREB becomes an essential ecological barrier and has an important strategic position, controlling its carbon emissions is crucial for green development in China [19,20].

Therefore, this study attempts to identify the spatio-temporal heterogeneity and key influencing factors of carbon emissions in the YREB of China. Specifically, local Moran’s I (LISA) was employed to verify the spatio-temporal distribution features of carbon emissions in the YREB of China. Moreover, an integrated method of the spatial Durbin model (SDM) and the extended Stochastic Impacts by Regression on Population Affluence and Technology model (STIRPAT), that is, the SDM-STIRPAT model, was employed to investigate the key influencing factors of carbon emissions in the YREB of China. Furthermore, integrated methods of Geographically and Temporally Weighted Regression (GTWR) and the extended STIRPAT model, that is, the GTWR-STIRPAT model are employed to investigate the spatio-temporal heterogeneity of carbon emissions in the YREB of China.

The rest of this study is organized as follows. Section 2 provides materials and methods, including the development process of the STIRPAT and variables selection, data sources, and models specification. Section 3 gives empirical results and analysis. Section 4 concludes the main findings, delivers the policy implications, and points out the research prospects.

## 2. Materials and Methods

### 2.1. Development Process of the STIRPAT and Variables Selection

The I = PAT formulation was first put forward in the 1970s to analyze the environmental stress of anthropomorphic factors [21,22]. Then, in the 1990s, Dietz and Rosa reformulated the IPAT model into a stochastic form (STIRPAT), which solved the limitation on the proportionate relationship between environmental pressure index and driving factors [23,24]. The IPAT model was mathematically defined as follows:(1)$$I=aP^{b}A^{c}T^{d}e$$
where *I* denotes the environmental press indicator, and represents carbon emissions in this study [25]. *P* denotes the population scale; *A* denotes the affluence; *T* denotes the technology; *a* is the coefficient of model; *b*, *c* and *d* represent the parameters to be estimated respectively; *e* denotes the error term.

In order to further detect the drivers of carbon emissions in the YREB, additional factors had been added to the formulation. Moreover, on the basis of previous studies, the STIRPAT model was formulated as the logarithmic form, which not only weakened the heteroskedasticity but also converted the model into linear form to facilitate estimation.
(2)$$\begin{array}{c} \ln I=a+b_{1}\ln P+b_{2}\ln A+b_{3}\ln T+b_{4}\ln SI+b_{5}\ln TI\\ +b_{6}\ln FAI+b_{7}\ln UR+b_{8}\ln OP+\ln e \end{array}$$
where *P*, *A*, *T*, *SI*, *TI*, *FAI*, *UR*, and *OP* denote population, affluence, technology, second industry, tertiary industry, fixed assets investment, urbanization, and openness, respectively. Specifically, population denotes the demographic influence; affluence and fixed assets investment represent the socioeconomic situation; technology denotes the technological progress; second industry and tertiary industry denotes the industrial structure; urbanization denotes the city condition; openness denotes the scale of foreign trade. The definition of the above variables is presented in Table 1.

Considering the shortage of direct statistical data, the carbon emissions can be derived from both direct and indirect sources [26]. The direct sources include fossil energies, such as coal, liquefied petroleum gas (LPG), natural gas, etc., while the indirect sources mainly include thermal power generation. The direct carbon emissions (*I^direct^*) can be calculated on the basis of the 2006 IPCC National Greenhouse Gas inventories as follows:(3)$$I^{direct}=\sum_{i}^{}E_{i}\times Cal_{i}\times Cc_{i}\times O_{i}\times \frac{44}{12}$$
where *Ei* denotes the consumption of various energies, including natural gas, liquefied petroleum gas (LPG), and raw coal; *Cal* refers to the net calorific value; *Cc* refers to carbon content; *Oi* refers to the carbon oxidation rate. Moreover, 44/12 denotes the coefficient of carbon transforming to carbon dioxide.

The indirect carbon emissions (*I*^indirect^) can be measured as follows:(4)$$I^{indirect}=Electricity\times F_{electricity}$$
where *electricity* refers to the electricity consumption; F_*electricity*_ refers to the CO_2_ emissions coefficient of indirect energy.

Therefore, the total carbon emissions can be demonstrated as:(5)$$I=I^{direct}+I^{indirect}$$

### 2.2. Data Resource

Scanning from 2006 to 2019, the panel data including carbon emissions and various independent variables are obtained from the China Energy Statistical Yearbook, China City Statistical Yearbook, China Regional Statistical Yearbook, and Statistical Yearbooks of individual provinces and cities. The descriptive statistics of the data are presented in Table 2.

### 2.3. Models Specification

#### 2.3.1. Spatial Auto-Dependence Model

It is widely known that global Moran’s I could merely identify the whole distribution characteristic of CO_2_ emissions, while the identification of the spatial auto-dependence between individual cities could be conducted by the local Moran’s *I* and illustrated by the LISA diagram [27]. Specifically, the formula of the local Moran’s *I* can be calculated as follows:(6)$$I_{i}=\frac{\left(x_{i}-\bar{x}\right)\sum_{j}\left[w_{ij}(x_{j}-\bar{x})\right]}{\sum_{i}{\left(x_{i}-\bar{x}\right)}^{2}/n}$$
where *i* and *j* denote two different cities, *x_i_* denotes carbon emissions in the city *i*, denotes the annual average carbon emissions, *W_ij_* denotes the 0–1 adjacency weight matrix in this study [28].
(7)$$W_{ij}=\{\begin{array}{l} 1,\;\mathrm{when}\;\mathrm{the}\;\mathrm{city}\;i\;\mathrm{and}\;j\;\mathrm{are}\;\mathrm{adjacent}\;\\ 0,\;\mathrm{when}\;i=j\;\mathrm{or}\;\mathrm{they}\;\mathrm{are}\;\mathrm{not}\;\mathrm{adjacent} \end{array}$$

Five spatial aggregation types are present in the LISA diagram, including “High-High” agglomeration, “Low-High” agglomeration, “High-Low” agglomeration, and “Low-Low” agglomeration, and not significant, respectively [29]. In particular, the “High-High” agglomeration and the “Low-High” agglomeration suggest that the sample city is positively associated with surrounding areas, that is, the spatial homogeneity of carbon emissions. While “High-High” agglomeration and “Low-High” agglomeration suggest that the sample city is negatively associated with surrounding areas, that is, the spatial heterogeneity of carbon emissions.

#### 2.3.2. Spatial Econometric Models

Traditional spatial regression models usually concern the space features, while temporal series analysis models only concern the trend features. Spatial panel models, nevertheless, focus on the temporal trend and the spatial distribution of the sample simultaneously, which mainly include the spatial lag model (SLM), the spatial error model (SEM), as well as the spatial Durbin model (SDM) model. Regarding the model comparison, the SDM model integrates the spatial spillover effects from explanatory variables and the lagged explained variable simultaneously, which can comprehensively capture the direct and indirect effects generated by various samples during the investigation period [30,31]. In this study, considering the potential existence of both the direct and indirect effects, the spatial Durbin model (SDM) under the space and time fixed effect was introduced to explore the relevance between carbon emissions and their driving factors in 56 cities in the YREB from 2006–2019, and the corresponding formula was taken as follows.
(8)$$\begin{array}{l} \ln I_{it}=\rho W\ln I_{it}+\beta_{1}\ln P+\beta_{2}\ln A+\beta_{3}\ln T+\beta_{4}\ln SI+\beta_{5}\ln TI+\beta_{6}\ln FAI\\ \;\;\;\;\;\;\;\;\;\;\;+\beta_{7}\ln UR+\beta_{8}\ln OP+\theta_{1}W\ln P+\theta_{2}W\ln A+\theta_{3}W\ln T+\theta_{4}W\ln SI\\ \;\;\;\;\;\;\;\;\;\;\;+\theta_{5}W\ln TI+\theta_{6}W\ln FAI+\theta_{7}W\ln UR+\theta_{8}W\ln OP+\mu_{i}+\tau_{i}+\varepsilon_{it} \end{array}$$
where *β* refers to the coefficient of direct effects, *θ* refers to the indirect coefficient of independent variables, while *ρ* refers to the indirect coefficient of the dependent variable; *t* denotes the *t-th* year and *i* denotes the city *t*; *W* denotes spatial weight matrix (equals to Equation (7)); *μ_i_* represents the space fixed effect, *τ_i_* represents the time fixed effect, and *ε_it_* represents the random error term. In addition, other parameters are equivalent to the meanings in the Formula (2).

Nevertheless, except for the explanatory variables’ impacts on CO_2_ emissions, the explained variable may also have indirect or spatial spillover impacts on itself. Therefore, the dynamic spatial Durbin model was also applied for further analysis, with the consideration of both temporal and spatial factors. Specifically, the dynamic spatial Durbin model is presented as follows:(9)$$\begin{array}{l} \ln I_{it}=\tau \ln I_{it-1}+\lambda W\ln I_{it-1}+\rho W\ln I_{it}+\beta_{1}\ln P+\beta_{2}\ln A+\beta_{3}\ln T+\beta_{4}\ln SI\\ \;\;\;\;\;\;\;\;\;\;\;+\beta_{5}\ln TI+\beta_{6}\ln FAI+\beta_{7}\ln UR+\beta_{8}\ln OP+\theta_{1}W\ln P+\theta_{2}W\ln A\\ \;\;\;\;\;\;\;\;\;\;\;+\theta_{3}W\ln T+\theta_{4}W\ln SI+\theta_{5}W\ln TI+\theta_{6}W\ln FAI+\theta_{7}W\ln UR\\ \;\;\;\;\;\;\;\;\;\;\;+\theta_{8}W\ln OP\mu_{i}+\tau_{i}+\varepsilon_{it} \end{array}$$
where *τ* denotes the temporal lag auto-regressive coefficient of the explained variable, and *λ* denotes the spatial lag auto-regressive coefficient of the explained variable. In addition, other parameters are equivalent to the meanings in the Formula (8).

#### 2.3.3. Geographically and Temporally Weighted Regression Model

In order to further analyze the spatial and temporal heterogeneity of the driving factors in the YREB, the Geographically and Temporally Weighted Regression (GTWR) model was applied in this research, which extends and refines the GWR model by incorporating arbitrary functions of the geo-location and time into the linear regression model [32]. Specifically, the GTWR model is described as follows:(10)$$Y_{i}=\beta_{0}\left(u_{i},v_{i},t_{i}\right)+\sum_{k=1}^{}\beta_{k}\left(u_{i},v_{i},t_{i}\right)X_{ik}+\varepsilon_{i}$$
where *Y_i_* denotes the logarithmic value of carbon emissions of city *i*, and *X_ik_* denotes the logarithm of each independent variable of city *i*; *β_k_* (*μ_i_*, *v_i_*, *t_i_*) denotes the regression coefficient of the *k-th* independent variable of city *i*, which is a function of spatio-temporal coordinates; *β*_0_ (*μ_i_*, *v_i_*, *t_i_*) denotes the spatio-temporal intercept term; denotes the error term and obeys the N (0,*σ^2^*) distribution.

To measure the intercept term and the independent variable coefficients, the Gaussian kernel function is used to calculate the spatio-temporal weight matrix.
(11)$$\hat{\beta }\left(u_{i},v_{i},t_{i}\right)={\left[X^{T}W\left(u_{i},v_{i},t_{i}\right)X\right]}^{-1}X^{T}W\left(u_{i},v_{i},t_{i}\right)Y$$

The spatial-temporal distance from point *i* to point *j* can be described as:(12)$$d_{ij}=\sqrt{\lambda \left[{\left(u_{i}-u_{j}\right)}^{2}+{\left(v_{i}-v_{j}\right)}^{2}\right]+\mu {\left(t_{i}-t_{j}\right)}^{2}}$$
where *λ* and *μ* represent scale variables to counterbalance the spatial effects and temporal effects, respectively. Since the units of measurement for time and space are often inconsistent, a direct and easy modeling approach used in this case is to incorporate the temporal and spatial distances into the spatio-temporal distance formulas. In particular, the space-time distance *d^ST^* is combined with the spatial distance *d^S^* and the temporal distance *d^T^* as follows:(13)$$d^{ST}=\lambda d^{S}+\mu d^{T}$$

It should be noted that the spatio-temporal weight matrix *W^ST^* is constructed by employing a decay function based on Euclidean distance and Gaussian distance as:(14)$$\begin{array}{l} {\left(d_{ij}^{ST}\right)}^{2}=\lambda \left[{\left(u_{i}-u_{j}\right)}^{2}+{\left(v_{i}-v_{j}\right)}^{2}\right]+\mu {\left(t_{i}-t_{j}\right)}^{2}\\ W^{ST}=\exp\left\{-\left(\frac{\lambda \left[{\left(u_{i}-u_{j}\right)}^{2}+{\left(v_{i}-v_{j}\right)}^{2}\right]+\mu {\left(t_{i}-t_{j}\right)}^{2}}{h_{ST}^{2}}\right)\right\}\\ \;\;\;\;\;\;\;\;\;=\exp\left\{-\left(\frac{{\left(u_{i}-u_{j}\right)}^{2}+{\left(v_{i}-v_{j}\right)}^{2}}{h_{S}^{2}}+\frac{{\left(t_{i}-t_{j}\right)}^{2}}{h_{T}^{2}}\right)\right\}\\ \;\;\;\;\;\;\;\;\;=\exp\left\{-\left(\frac{{\left(d_{ij}^{S}\right)}^{2}}{h_{S}^{2}}+\frac{{\left(d_{ij}^{T}\right)}^{2}}{h_{T}^{2}}\right)\right\}\\ \;\;\;\;\;\;\;\;\;=\exp\left\{-\frac{{\left(d_{ij}^{s}\right)}^{2}}{h_{S}^{2}}\right\}\times \exp\left\{-\frac{{\left(d_{ij}^{T}\right)}^{2}}{h_{T}^{2}}\right\}\\ \;\;\;\;\;\;\;\;\;=W^{S}\times W^{T} \end{array}$$
where *t_i_* and *t_j_* denote the observation times at positions *i* and *j*, respectively. *h^ST^*, *h^S^*, *h^T^* denote the specifications of the spatio-temporal bandwidth, spatial bandwidth, and temporal bandwidth, respectively. Meanwhile, this study adopts *AIC* and *R*^2^ to determine the fitness of this model.

## 3. Empirical Analysis

### 3.1. Spatial Auto-Correlation of Carbon Emissions in the YREB

In order to visualize the spatial distribution characteristics of the carbon emissions in YREB, this paper employs the LISA diagram (Figure 1) drawn by Geoda software to present agglomerations. Figure 1 demonstrates the clustering features of urban carbon emissions in the YREB under the 5% significance level. In general, both the High-High and Low-Low clusters were dominant in the sample period, which suggests that there is remarkable spatial positive dependence in these regions. Specifically, the High-High clusters were mainly concentrated in the Shandong Peninsula, such as Dongying, Jinan, linyi, Zibo, and Binzhou. However, the High-High clusters spread to Hohhot and Baotou in 2019, one possible reason is that local officials tend to glorify their political performance by focusing on economic growth rather than pollution reduction, and this leads to the “race to the bottom” in the High-High clusters [33]. While the Low-Low agglomerations were mainly distributed in the upstream and midstream cities, including Qingyang, Yan’an, Yulin, Linfen, Pingliang, Tianshui, Tongchuan, Longnan, Zhongwei, Baiyin, and Dingxi, highlighting the spatial heterogeneity of carbon emissions in the YREB.

Moreover, the scale of the High-High and Low-Low agglomerations expanded from 2006 to 2019, indicating the increasing trend of spatial dependence between neighboring areas. In addition, both the High-High and Low-Low clusters are relatively stable, indicating the “local club effects”. As for the High-Low agglomerations, Lanzhou was located in the surrounding of the Low-Low agglomerations, while Dezhou and Bayannur were located in the surrounding of the High-High agglomerations. Therefore, to avoid the expansion of high carbon pollution in the YREB, overall coordination rather than local incentives is more necessary.

### 3.2. Identifying Key Influencing Forces of Carbon Emissions in the YREB

#### 3.2.1. Static Spatial Panel Analysis

Before estimation, a series of diagnostic tests are conducted to verify the fitness of the SDM, and they are reported in Table 3. For instance, the LM test indicates that the original hypothesis of “no spatial autocorrelation” is refused at the 1% significance level; the Hausman test based on the SDM indicates the random effect model is unavailable at the 1% significance level. In addition, the combined LR tests suggest that the SAR and the SEM cannot be nested in the SDM at the 5% significance level and the 1% significance level, respectively, and the time fixed effect SDM and space fixed effect SDM cannot be nested in the space-and-time fixed effect SDM at the 1% significance level. Furthermore, the R_squared of the SDM under the space-and-time fixed effect is 0.943 (Table 4), suggesting this model has a good fitting effect. Therefore, after a series of diagnostic tests, the SDM under the space-and-time fixed effect is identified as reasonable for this experiment.

Due to the existence of spatial feedback effects among various variables, the direct and indirect coefficients of the variables are not fully consistent with the effect coefficients of the variables. First, from the transmission mechanisms, the direct coefficient of ln*P* is remarkably positive at the 1% significance level, which suggests that the expansion of population scale would exaggerate local carbon emissions. This is due to the higher population density of YREB, in other words, the incompatibility between the extent of population concentration and environmental carrying capacity leads to the increase in local carbon emissions [34]. However, the indirect and total effects coefficients of ln*P* have not been approved by the significance test, indicating that the spillover effects of the population have not been exerted.

Second, the direct coefficient of ln*A* is highly positive at the 1% significance level, demonstrating that there is a positive correlation between affluence and local carbon emissions in the YREB. According to the Carbon Kuznets curve (CKC) theory, the carbon emissions resulting from the economic growth in the YREB have not yet reached the inflection point, which is due to the agglomeration effects and positive externality of economic growth have not been realized in the YREB [35]. However, the indirect coefficient of ln*A* has not passed the significance test, indicating that the spatial spillover effect of affluence has not been exerted.

Third, the direct coefficient of ln*T* is positive at the 1% significance level, indicating that the technological process has increased local carbon emissions in the YREB. However, the indirect coefficient of ln*T* has not passed the significance test, indicating that the spatial spillover effect of affluence has not been exerted. This is because of backward and inefficient production methods, energy consumption and carbon emissions per unit GDP have risen significantly in the YREB.

Fourth, the direct coefficient of ln*SI* is positive but insignificant, hence it is difficult to define the impact of the secondary industry on carbon emissions in the YREB. While the direct coefficient of ln*TI* is positive at the 10% significance level, which demonstrates that the increase in the tertiary industry promotes carbon pollution, which is related to the massive carbon emissions generated by transportation and postal industries. However, the indirect coefficient of ln*SI* and ln*TI* have not passed the significance test, indicating that the spatial spillover effects of the secondary industry and tertiary industry on carbon emissions are relatively weak.

Fifth, the direct coefficient of ln*FAI* is positive but insignificant, that is, no clear proof supports the positive impact of fixed assessment investment on carbon emissions in the YREB. In addition, the indirect and total coefficients of ln*FAI* have not passed the significance test, indicating that the spatial spillover effect of fixed asset investment has not been exerted.

Sixth, the direct coefficient of ln*UR* is positive at the 5% significance level, indicating that urbanization has increased local carbon emissions, one possible reason is that with the increase in urban population, the rapidly developing infrastructure leads to excessive energy consumption and enhanced emissions; therefore, urbanization can greatly contribute to carbon emissions in transportation and building sectors. On the other hand, the indirect and total coefficients of ln*UR* all failed the significance test, implying the spatial effect of urbanization on carbon emissions is relatively weak in the YREB.

Seventh, the direct coefficient of ln*OP* plays a relatively minor but negative effect at the 1% significance level, indicating that foreign trade has hindered local carbon emissions in the YREB. Meanwhile, both the indirect and total coefficients of ln*OP* are negative at the 1% significance level, indicating that the development of foreign trade in neighboring cities has contributed to carbon emissions reduction in the particular city; therefore, the “Pollution Halo” hypothesis rather than the “Pollution Heaven” hypothesis is supported in the YREB, which means that the development of foreign trade introduces clean technologies to YREB and thus reduces carbon emissions to a certain degree [36].

Last but not the least, the value of spatial rho is remarkably positive at the 1% significance level, which demonstrates that the spatial spillover effect of carbon emissions among different cities is supported in the YREB.

#### 3.2.2. Dynamic Spatial Panel Analysis

The estimation results of the dynamic SDM under the space-and-time fixed effect are presented in Table 5. Compared with the static estimation results reported in Table 4, several novel findings can be drawn here.

First, the temporal lag coefficient of ln*I* is positive at the 1% significance level, and the spatial lag coefficient of ln*I* is positive at the 5% significance level, implying that the previous carbon emissions in the particular city would increase the present carbon emissions in both local and surrounding cities. Thus, the carbon emissions have a snowball effect, consistent with previous speculations.

Second, the direct coefficients of ln*P* imposed by the short-term and the long-term are positive at the 1% significance level, while the value of the coefficient imposed by the long-term is greater than it imposed by the short-term, demonstrating that the direct effect of the population over the long term is stronger than it over the short term. Similarly, the direct coefficients of ln*A* and ln*T* in the short-run and the long-run are positive at the 1% significance level, while the value of the coefficients imposed in the long-run is both greater than those imposed in the short-run, demonstrating that the direct effect of influence and technology, in the long run, is stronger than in the short run. Moreover, the direct coefficients of ln*OP* imposed by the short-term and the long-term are negative at the 1% significance level, while the absolute value of the coefficient imposed by the long-term is greater than it imposed by the short-term, indicating that openness has negatively enhanced carbon emissions in the YREB. However, the indirect coefficients of all other independent variables failed the significance test, showing that these indirect effects have failed to exert a stronger effect on carbon emissions in the long-term.

Furthermore, the total coefficients of ln*A* and ln*T* are positive in the long-term and in the short-term, and the value of the coefficients is greater in the long-term than that in the short-term, which highlights the establishment of spatial decomposition in estimation. In addition, the total coefficients of ln*OP* imposed by the short-term and the long-term are both negative at the 1% significance level, and the total effect of it is more powerful in the longer run, indicating that there is a temporal lag effect of this factor.

Last but not least, compared with the absolute value of the coefficients including second industry, tertiary industry, fixed assets investment, urbanization, and openness, the three traditional factors including population, affluence, and technology are identified as the key influencing factors of carbon emission in the YREB under the empirical framework of DSDM-STIRPAT. However, since openness does not belong to the key influencing factors, its negative effect on reducing carbon emissions in the YREB also highlights the importance and necessity of an open economy.

### 3.3. Spatio-Temporal Heterogeneity Analysis

To further analyze the spatio-temporal heterogeneity of the key influencing factors, the corresponding results of the GTWR model are mapped in Figure 2, Figure 3 and Figure 4. First, the value of AIC (Akaike information criterion) is −851, and R_squared is 0.9912, indicating the GTWR model fits well. In addition, most coefficients of ln*P*, ln*A*, ln*T* have passed the significance test, implying the GTWR model is valid. As for the remaining five variables, the results in the sample cities do not fully pass the significance test, thus the fact that they are not the key influencing factors is proved once again.

In terms of population (Figure 2), generally speaking, the expansion of population scale increases carbon emissions in the YREB, that is, all sample cities would generate carbon emissions owing to the increasing population. However, the heterogeneous influence of the population on carbon emissions in space should not be ignored. Specifically, the effect coefficients in Taiyuan, Yangquan, Jinan, Dongying, Zibo, Jining, Linyi, Taian, Liaocheng, Luoyang, Sanmenxia, and Xining are relatively high, while the effect coefficients in Wuhai, Bayannur, Ordos, Baoji, Xianyang, Tianshui, Baiyin, and Shizuishan are relatively low.

Furthermore, the influence of the population on promoting carbon emissions is heterogeneous in different periods. For instance, the cities performing a stronger promoting effect over time are mostly concentrated in Linfen, Lvliang, Yuncheng, Ulanqab, while the cities performing a weaker driving effect over time are distributed in Jinan, Zibo, Binzhou, Dongying, Dezhou, Zhengzhou, Xinyang, and Xinin.

With respect to affluence (Figure 3), the impact of economic growth is similar to population, since the coefficients are positive for all sample cities during the sample period. Similarly, the coefficients of different cities also vary spatially. Cities with strong driving forces from affluence include Ordos, Jinan, Zibo, Jining, Linyi, Tai’an, Liaocheng, Heze, Dezhou, Binzhou, Dongying, Jiaozuo, Sanmenxia, and Anyang, with fast-growing economy and good industrial base. Cities with weak driving forces from affluence include Lanzhou, Baiyin, Dingxi, and Longnan, which are located in the west region, with a relatively backward economy, low urbanization and less energy consumption.

In addition, the coefficients of different cities also vary temporally. For instance, the cities with increasing effects of influence on carbon emissions during the sample period include Xining, Shizuishan, Baoji, Linfen, Lvliang, Tianshui, and Jinzhong; while the cities enjoying decreasing effects of affluence on carbon emissions during the sample period include Weinan, Tongchuan, Sanmenxia, Xi’an, Luoyang, Kaifeng, and Puyang.

In the case of technology (Figure 4), all sample cities show promoting effects on carbon emissions in the YREB. Specifically, the cities enjoying relatively high driving forces include Weinan, Xianyang, Tongchuan, Xi’an, Sanmenxia, Zibo, Dongying, and Luoyang; while the cities with relatively weak driving forces include Guyuan, Zhongwei, Wuzhong, Baiyin, Lanzhou, Taiyuan, Yangquan, and Jinzhong.

Furthermore, the coefficients of different cities also vary temporally. For example, Tongchuan, Xi’an, Sanmenxia, Xianyang, and Weinan show an increasing trend, while Yulin, Zhongwei, Pingliang, Dingxi, Tianshui, Lanzhou, and Baiyin show a decreasing trend oppositely.

In short, the spatio-temporal heterogeneity of the key influencing factors is proved, and the importance of suiting measures to local conditions in terms of space and time involved should be highlighted in further research.

## 4. Conclusions, Policy Implications, and Research Prospects

### 4.1. Conclusions

The purpose of this study is to identify the spatio-temporal heterogeneity of carbon emissions and their driving forces in the YREB of China. Specifically, this study employed the LISA diagram to identify the spatial distribution characteristics of carbon emissions in the YREB. In addition, this paper used both the spatial Durbin model (SDM) and the dynamic spatial Durbin model (DSDM) under the space-and-time fixed effect to investigate the key influencing factors. Third, the spatio-temporal heterogeneity was further investigated based on the application of the GTWR model. The conclusions can be summarized as follows:(1)In terms of spatial characteristics, in the carbon emissions in the YREB significant clustering characteristics appeared. The Low-Low clusters were mainly distributed in the midstream of the YREB, while the High-High clusters were mainly located in the Shandong Peninsula, and compared with other cities, the carbon emissions of Jinan, Linyi, and Dongying were greater, which is related to the fact that the economic growth of Shandong Peninsula mainly benefited from heavy industries with high-energy consumption and high emissions. Despite the rapid economic growth in the Shandong Peninsula, the deterioration of the ecological environment is accompanied by increased carbon emissions [37]. In addition, the High-High clusters exhibited an apparent high-carbon spillover effect, specifically reflected in the spatial expansion of the high-carbon emission areas over time, and the transformation of the Low-High clusters and insignificant clusters into High-High clusters, which should be examined by local governments. Whereas the low-carbon lock-in effect mainly appeared in the Low-Low clusters, such as Yan’an, Pingliang, Tianshui, Tongchuan, Longnan, Zhongwei, Baiyin, and Dingxi, where the spatial pattern is more solid.(2)The estimation results of the spatial panel analysis demonstrated that the carbon emissions showed a distinct spatial lag effect and temporal lag effect. In addition, population, affluence, technology, tertiary industry, and urbanization possessed significantly direct effects, while only openness played a negative direct effect and indirect effect. Meanwhile, affluence and technology all had remarkably positive total effects on carbon emissions, while openness exerted a negative total effect on carbon emissions. The economic development model of the YREB which relies only on inputs of production supplies including capital and labor to enhance outputs could result in an increment in carbon emissions. While the tertiary industrial structure and technology contribute to the suppression of carbon emissions in the YREB; the restraint effect is so weakened that it cannot offset the growth of carbon emissions caused by their stimulating factors. Moreover, the result that urbanization is one of the contributors to promoting carbon emissions is consistent with previous studies [38]. Urbanization has a huge clustering effect on economic development, and there are intensive construction activities, extensive transportation systems and high population density in urban areas, which will increase energy consumption and carbon emissions. Thus, the governments of the YREB ought to change their economic development models to intensive production and realize the de-coupling association between economic growth and carbon emissions [39]. In general, the effects in the long-term are stronger than those in the short-term, which highlights the existence of the circular cumulative effect and the rationality of the findings in this study.(3)The influences of different variables on carbon emissions exhibit remarkable spatio-temporal heterogeneity. Even in the same basin, there is significant spatio-temporal heterogeneity among key influencing factors including affluence, population, and technology, for example, the three main drivers are stronger in the middle and downstream cities and relatively weaker in the upstream cities of the YREB; simultaneously, in the middle and downstream cities of the YREB, the influence of technology is increasing but the influence of affluence is decreasing. This heterogeneity seems to be driven by the segmentation of regional economic and social development status, certificating that geographical location and temporal factors are essential when investigating the driving factors of carbon emissions.

### 4.2. Policy Implications

In the light of the above conclusions, the integrated policy implications are stated as follows:

First, considering the spatial agglomeration of carbon emissions, the cities in the High-High agglomeration, such as the cities in the Shandong peninsula, should take a pioneering act in environmental regulation to curb the high-carbon spillover effect, and the potential “race to bottom” effect ought to be controlled while the “race to top” effect ought to be promoted according to the “common but different” guidelines of pollution management [40]. In addition, cities in the Guanzhong City Cluster should strive to promote technology innovation and clear production, to play radiation effects on surrounding High-Low clusters, such as Lanzhou. Similarly, cities in the Low-High cluster, such as Bayannur and Ulanqab, should leverage spatial spillover effects and improve the carbon lock-in situation, with industrial structure optimization and openness trade expansion.

Second, as affluence, population and technology contribute significantly to carbon emissions, governments should adopt a high-quality development strategy and aim to pursue a low-carbon economy to decouple the economy from carbon emissions [41]. Furthermore, the adjustment of population inflow and the introduction of advanced technology should be combined with the national strategy of “One Belt, One Road” [42]. In addition, to reduce the carbon emissions in the progress of technological and economic development, clear producing technology ought to be a priority for technological innovation strategy. To realize these goals, the governments of YREB need to change the existing extensive economic model to an intensive economic model, specifically, governments should apply economic stimulus measures more efficiently, such as encouraging low-carbon companies to research and develop new technologies and lessening their trouble in obtaining loans [43]. Simultaneously, considering the “pollution heaven” effect of foreign trade in the YREB, enhancing the reduction effect of openness on carbon emissions is essential for both the local and surrounding areas in the YREB, and as the urbanization rate is another contributor to carbon emissions; it is recommended to enhance the efficiency of the urbanization efforts in the YREB. Since the tertiary industry is also a factor to increased carbon emissions in the YREB, the policies should focus on restructuring industries, particularly promoting the eco-friendly tertiary industries including commerce, services, tourism and retail.

Third, the temporal lag effect and spatial spillover effect should be taken into account when formulating policies, and the efficient management of carbon emissions depends on the joint efforts of local governments and the sustainability of policy implementation. Furthermore, to release the power of industrial upgrading on reducing carbon emissions in the long-term, the integration and updating of industrial structure should be put into the performance evaluation of local government [44,45]. Moreover, to strengthen the spatial spillover effects of green economy policies, a regional collaboration system based on geographical relevance should be developed, designed to establish a long-term functioning mechanism.

Last but not least, to respond to the demand for spatio-temporal heterogeneity, the formulation of carbon policies should be conducted according to local conditions, and cross-regional cooperation among local governments should be promoted [46]. Meanwhile, the transformation of the development model from energy oriented, cleaner production and technology innovation of energy-intensive industries ought to be optimized. Furthermore, local authorities must foster high-tech and eco-friendly industries, to reduce energy consumption and curb carbon pollution based on their resource endowment. In addition, to reduce the space of vicious competition, the realization of carbon emission reduction and energy saving at the national level should be combined with the unified planning of the central government’s top-level design.

### 4.3. Research Prospects

Although this study has comprehensively investigated the spatio-temporal heterogeneity and key influencing factors of carbon emissions in the YREB of China, some potential directions still deserve further exploration. For instance, the natural conditions including smoke, dust, temperature, and humidity also had a significant impact on carbon emissions, further research may include them under the availability of data. In addition, a comparison with the spatio-temporal heterogeneity and key influencing factors of carbon emissions in the Yangtze River Economic Belt and the Yellow River Economic Belt may lead to more interesting findings, and more spatial econometric models, such as the Geographical Detector and the Standard Deviation Ellipse, which may be employed in the near future.

## Figures and Tables

**Figure 1 ijerph-19-04185-f001:**
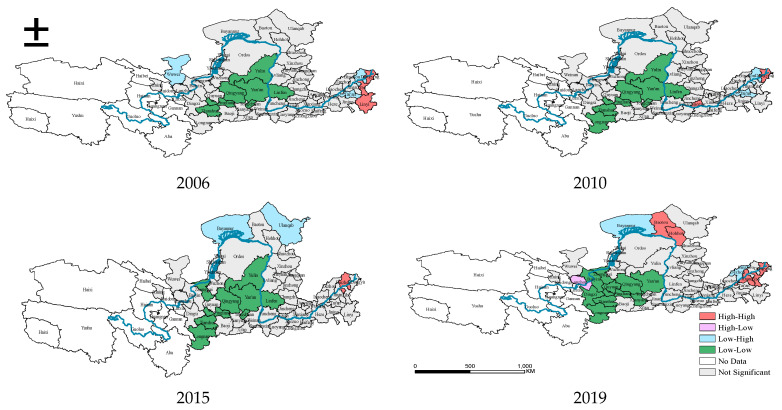
Spatial clustering of carbon emissions.

**Figure 2 ijerph-19-04185-f002:**
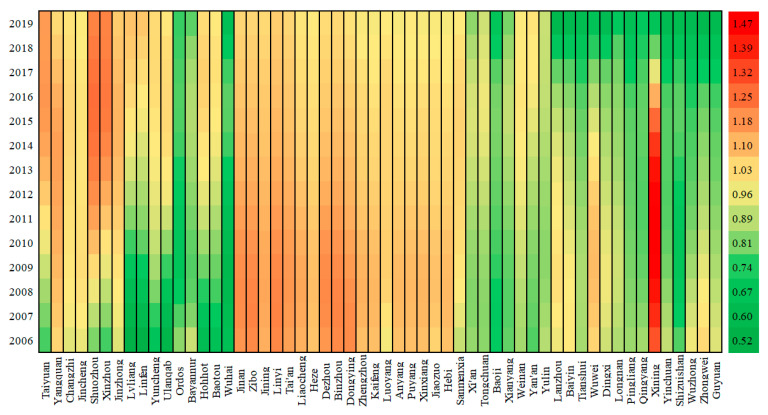
Heat Map of population coefficient.

**Figure 3 ijerph-19-04185-f003:**
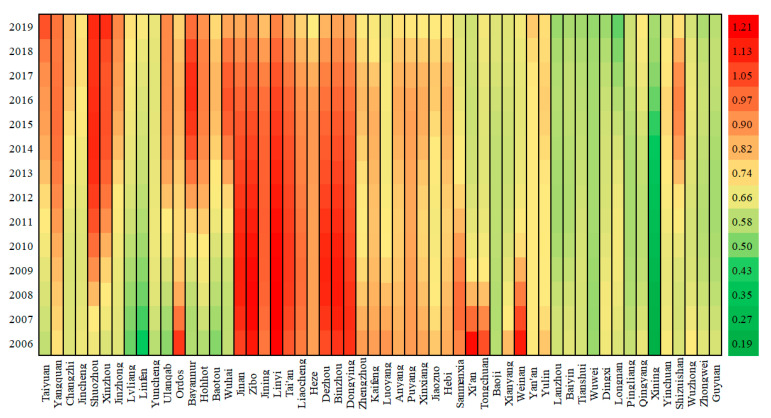
Heat Map of affluence coefficient.

**Figure 4 ijerph-19-04185-f004:**
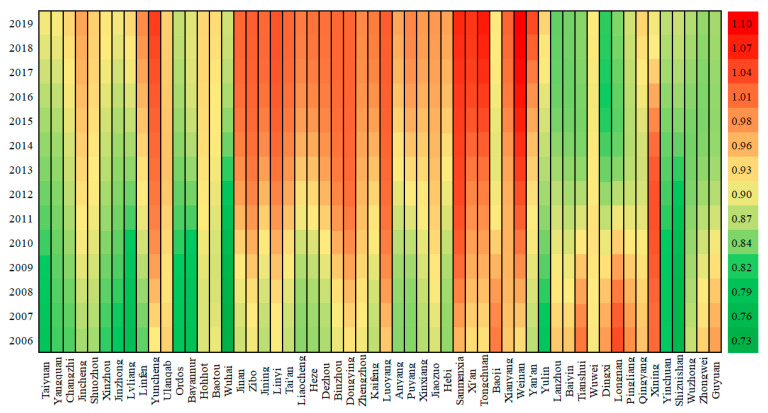
Heat Map of technology coefficient.

**Table 1 ijerph-19-04185-t001:** Definition of the variables.

Variables	Definition	Unit	Symbol
CO_2_ emission	Energy-related carbon emission accounting	10^4^ Ton	*I*
Population	End year total population	10^4^ Persons	*P*
Affluence	GDP per capita	10^4^ Yuan	*A*
Technology	Energy consumption per million GDP	Ton/10^4^ Yuan	*T*
Second Industry	The proportion of secondary industry output in GDP	%	*SI*
Tertiary Industry	The proportion of tertiary industry output in GDP	%	*TI*
Fixed Assets Investment	Total fixed asset investment	10^8^ Yuan	*FAI*
Urbanization	Ratio of urban population to total population	%	*UR*
Openness	The share of total imports and exports to GDP	%	*OP*

**Table 2 ijerph-19-04185-t002:** Descriptive Statistics.

Variable	Obs.	Mean	S.D.	Min	Max
CO_2_ emission (*I*)	784	930.069	965.348	14.865	4496.780
Population (*P*)	784	375.482	235.013	48.060	1083.800
Affluence (*A*)	784	3.464	2.845	0.395	16.424
Technology (*T*)	784	1.640	1.820	0.090	9.760
Second Industry (*SI*)	784	50.883	11.766	20.660	74.690
Tertiary Industry (*TI*)	784	38.674	10.373	18.260	65.420
Fixed Assets Investment (*FAI*)	784	5506.764	5533.943	268.830	29,515.300
Urbanization (*UR*)	784	48.699	17.409	12.377	94.544
Openness (*OP*)	784	8.783	10.616	0.045	54.263

**Table 3 ijerph-19-04185-t003:** Diagnostic tests.

Test	Statistic	Test	Statistic
Moran’s I-error	3.467 ***	Hausman	53.83 ***
LM-error	10.968 ***	LR-test(Assumption: sar nested in sdm)	19.74 **
Robust LM-error	15.503 ***	LR-test(Assumption: sem nested in sdm)	40.47 ***
LM-lag	7.780 ***	LR-test(Assumption: sdm_time nested in sdm_both)	718.78 ***
Robust LM-lag	12.315 ***	LR-test(Assumption: sdm_ind nested in sdm_both)	85.32 ***

Note: t statistics in parentheses; ** *p* < 0.05, *** *p* < 0.01.

**Table 4 ijerph-19-04185-t004:** Estimation results based on the SDM.

Variables	Main	W·X	Direct Effect	Indirect Effect	Total Effect
ln*P*	0.725 ***	−0.119	0.734 ***	−0.003	0.732
	(3.022)	(−0.371)	(2.958)	(−0.007)	(1.545)
ln*A*	0.939 ***	−0.243 **	0.934 ***	−0.122	0.813 ***
	(9.590)	(−1.977)	(9.970)	(−0.963)	(5.786)
ln*T*	0.917 ***	−0.107 *	0.920 ***	0.008	0.928 ***
	(28.239)	(−1.859)	(29.454)	(0.234)	(16.968)
ln*SI*	0.158	−0.096	0.152	−0.102	0.050
	(1.222)	(−0.351)	(1.186)	(−0.351)	(0.147)
ln*TI*	0.144 *	−0.168	0.138 *	−0.179	−0.040
	(1.869)	(−0.926)	(1.913)	(−0.866)	(−0.191)
ln*FAI*	0.074	0.061	0.073	0.073	0.146
	(1.008)	(0.690)	(1.050)	(0.781)	(1.319)
ln*UR*	0.096 **	−0.016	0.095 **	−0.009	0.087
	(2.561)	(−0.244)	(2.559)	(−0.129)	(1.093)
ln*OP*	−0.026 ***	−0.090 ***	−0.029 ***	−0.102 ***	−0.131 ***
	(−2.628)	(−3.319)	(−2.950)	(−3.381)	(−4.048)
Spatial rho	0.122 **				
	(2.523)				
sigma2_e	0.013 ***				
	(19.556)				
Observations	784				
R-squared	0.943				

Note: t statistics in parentheses; * *p* < 0.1, ** *p* < 0.05, *** *p* < 0.01.

**Table 5 ijerph-19-04185-t005:** Estimation results based on the dynamic SDM.

Variables	Main	W·X	Short-Term			Long-Term		
			Direct Effect	Indirect Effect	Total Effect	Direct Effect	Indirect Effect	Total Effect
ln*I_t-1_*	0.116 ***							
	(3.440)							
W·ln*I_t-1_*	0.064 **							
	(2.028)							
ln*P*	0.626 ***	−0.175	0.645 ***	−0.132	0.512	0.729 ***	−0.092	0.636
	(2.826)	(−0.550)	(2.992)	(−0.404)	(1.161)	(2.955)	(−0.226)	(1.160)
ln*A*	0.846 ***	−0.262 **	0.843 ***	−0.206	0.637 ***	0.952 ***	−0.161	0.791 ***
	(7.472)	(−2.061)	(7.219)	(−1.535)	(3.587)	(7.175)	(−0.965)	(3.537)
ln*T*	0.847 ***	−0.115 **	0.846 ***	−0.058	0.788 ***	0.958 ***	0.020	0.978 ***
	(18.017)	(−2.248)	(18.046)	(−1.418)	(9.903)	(17.768)	(0.360)	(9.603)
ln*SI*	0.104	−0.075	0.106	−0.086	0.020	0.118	−0.095	0.023
	(0.904)	(−0.291)	(0.896)	(−0.320)	(0.064)	(0.874)	(−0.286)	(0.061)
ln*TI*	0.068	−0.072	0.070	−0.083	−0.014	0.077	−0.094	−0.017
	(0.928)	(−0.422)	(0.949)	(−0.480)	(−0.077)	(0.933)	(−0.447)	(−0.077)
ln*FAI*	0.086	0.081	0.088	0.079	0.168	0.102	0.106	0.208
	(1.123)	(0.959)	(1.166)	(0.905)	(1.605)	(1.190)	(0.999)	(1.606)
ln*UR*	0.058 *	−0.040	0.061 *	−0.035	0.026	0.068 *	−0.037	0.032
	(1.686)	(−0.724)	(1.791)	(−0.607)	(0.387)	(1.773)	(−0.516)	(0.385)
ln*OP*	−0.027 ***	−0.078 ***	−0.028 ***	−0.084 ***	−0.112 ***	−0.033 ***	−0.106 ***	−0.139 ***
	(−2.628)	(−3.058)	(−2.903)	(−3.303)	(−3.894)	(−3.018)	(−3.391)	(−3.891)
Spatial rho	0.070							
	(1.359)							
sigma2_e	0.012 ***							
	(5.420)							
Observations	728							
R-squared	0.952							

Note: t statistics in parentheses; * *p* < 0.1, ** *p* < 0.05, *** *p* < 0.01.

## Data Availability

The data used to support the findings of this study are available from the corresponding author upon request.

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
