# Peer review of "Spatio-Temporal Heterogeneity of Carbon Emissions and Its Key Influencing Factors in the Yellow River Economic Belt of China from 2006 to 2019"

_ijerph, 2022, doi:10.3390/ijerph19074185_

Round 1

Reviewer 1 Report

I congratulate the authors on this insightful and well-empirically researched article. Meanwhile, I will suggest that the authors include suggestion(s) for further studies.

Author Response

Although this study has comprehensively investigated the spatio-temporal heterogeneity and key influencing factors of carbon emissions in the YREB of China, some potential directions still deserve further exploration. For instance, the natural conditions including smoke, dust, temperature, and humidity also had a significant impact on carbon emissions, further research may include them under the availability of data. In addition, a comparison with the spatio-temporal heterogeneity and key influencing factors of carbon emissions in the Yangtze River Economic Belt and the Yellow River Economic Belt may lead to more interesting findings, and more spatial econometric models such as the Geographical Detector and the Standard Deviation Ellipse may be employed in the near future.

Reviewer 2 Report

After minor revision the manuscript will be ready for publication.

1. Improve the abstract

2. The results section is missing from the manuscript.

3. The references seem few for a scientific article. I suggest adding more.

4. The conclusions are based on public policy, but the title, methods, and results of the article do not include them. Could the authors clarify this or integrate it into the manuscript.

Author Response

Reply:Thank you very much for your suggestions. We have made further additions in the abstract, results and conclusions. Such as “The economic development model of the YREB which relies only on inputs of production supplies including capital and labor to enhance outputs, could result in the increment of carbon emissions. While the tertiary industrial structure and the technology contribute to the suppression of carbon emissions in the YREB, however, their restrain effect is so weakening that cannot offset the growth of carbon emissions caused by their stimulating factors. Moreover, the result that urbanization is one of the contributors of promoting carbon emissions is consistent with previous studies [38]. Urbanization has a huge clustering effect on economic development, and there are intensive construction activities, extensive transportation systems and high population density in urban areas, which will increase energy consumption and carbon emissions. Thus the governments of the YREB ought to change its economic development model to intensive production and realize the de-coupling association between economic growth and carbon emissions [39].”

“Third, the temporal lag effect and spatial spillover effect should be taken into account when formulating policies, and the efficient management of carbon emissions depends on the joint efforts of local governments and the sustainability of policies implementation. Furthermore, to release the power of industrial upgrading on reducing carbon emissions in the long-term, the integration and updating of industrial structure should be put in the performance evaluation of local government [44-45]. And then in order to strengthen the spatial spillover effects of green economy policies, a regional collaboration system on the basis of geographical relevance should be developed, designed to establish a long-term functioning mechanism.”

At the same time, we have adopted your friendly suggestions and added more scientific references as follows:

  1.  Chen, H. X.; Zhang, X. L.; Wu, R. W.; Cai, T. Y. Revisiting the environmental Kuznets curve for city-level CO2 emissions: based on corrected NPP-VIIRS nighttime light data in China.J. Clean. Prod.2020, 268,121575.
  2. Yang, H.;Lu, Z. N.; Shi, X. P.; Muhammad, S.; Cao, Y. How well has economic strategy changed CO2 emissions? Evidence from China's largest emission province. Sci. Total Environ. 2021, 774, 146575.
  3. Shuai, C.;Shen, L.; Jiao, L.; Wu, Y.; Tan, Y. Identifying key impact factors on carbon emission: evidences from panel and time-series data of 125 countries from 1990 to 2011. Applied Energy. 2017, 187, 310-325.
  4. Gao, C. C.; Ge, H. Q.; Lu, Y. Y.; Wang, W. J.; Zhang, Y. J. Decoupling of provincial energy-related CO2 emissions from economic growth in China and its convergence from 1995 to 2017. J. Clean. Prod.2020, 297, 126627.
  5. Shen, L.; Shuai, C.; Jiao, L.; Tan, Y.; Song, X. Dynamic sustainability performance during urbanization process between BRICS countries. Habitat International, 2017, 60, 19-33.
  6. Du, J. L.;Zhang, Y. F. Does One Belt One Road initiative promote Chinese overseas direct investment? China Economic Review.2017, 47, 189-205.
  7. Liu, Y.;Yang, D. W.; Xu, H. Z. Factors Influencing Consumer Willingness to Pay for Low-Carbon Products: A Simulation Study in China. Bus. Strategy Environ. 2017, 26(7), 972-984.

Thank you very much for your careful review and precious comments to refine our manuscript once again. We hope the current version has resolved all the questions and look forward to your feedback. Please feel free to contact me at the address below if you have any questions.

Best regards,

Jingxue Zhang, Yanchao Feng*, Ziyi Zhu

Business School, Zhengzhou University, Zhengzhou 450001, PR China

* Correspondence: [email protected]; Tel.: +86-150-0218-2995

Reviewer 3 Report

The study was well conducted but what is missing and significantly lowers the quality of the article is lack of a discussion of the results. The results are presented only as statistical description of the estimations. There need to be added at least several paragraphs on the meaning of these estimations and a comparison with similar studies.

Author Response

Reply:Thank you very much for your suggestions. Based on your suggestions, we have further added the description for discussing the results, such as “This is due to the higher population density of YREB, in other words, the incompatibility between the extent of population concentration and environmental carrying capacity leads to the increase of local carbon emissions [34].”

“According to the Carbon Kuznets curve (CKC) theory, the carbon emissions resulted from the economy growth in the YREB have not yet reached the inflection point, which is due to the agglomeration effects and positive externality of economic growth haven’t been realized in the YREB [35].”

“This is because of the backward and inefficient production methods, energy consumption and carbon emissions per unit GDP have risen significantly in the YREB.” 

Thank you very much for your careful review and precious comments to refine our manuscript once again. We hope the current version has resolved all the questions and look forward to your feedback. Please feel free to contact me at the address below if you have any questions.

Best regards,

Jingxue Zhang, Yanchao Feng*, Ziyi Zhu

Business School, Zhengzhou University, Zhengzhou 450001, PR China

* Correspondence: [email protected]; Tel.: +86-150-0218-2995

Reviewer 4 Report

The article is interesting and confirms the importance of taking up this topic. The diligence of the authors for so many models and for the results developed should be appreciated. The authors should also be commended for the clarity of the description of the models. The article can be accepted but I suggest linguistic correction because there are minor linguistic errors, for instance: " an integrated methods". 

Author Response

Reply: Thank you very much for your careful review and precious comments to refine our manuscript once again. We hope the current version has resolved all the questions and look forward to your feedback. Please feel free to contact me at the address below if you have any questions.

Best regards,

Jingxue Zhang, Yanchao Feng*, Ziyi Zhu

Business School, Zhengzhou University, Zhengzhou 450001, PR China

* Correspondence: [email protected]; Tel.: +86-150-0218-2995

Reviewer 5 Report

This is an interesting article analyzing the comprehensively empirical framework to identify the spatio-temporal heterogeneity of carbon emissions and its driving forces in the YREB of China from 2006 to 2019. 

The paper’s title match its content. The issue presented in the paper have practical applications. The research topic presented clearly, aim of article clearly specified and realized. The article have a logical layout. The language of article correct. The paper’s conclusions follow logically from the development of the argument. The text adequately illustrated (tables and figures). Statistical analysis is sufficient and appropriate.

Two suggestions for Authors: 

  • The article lacks a solid discussion of the results and comparisons with other studies. This should be completed. 
  • The article requires a minor editorial correction (e.g. CO2 - CO2 etc.).

Author Response

Reply:Thank you very much for your suggestions. We have refined the discussion of the results and compared the results with other studies. such as “The economic development model of the YREB which relies only on inputs of production supplies including capital and labor to enhance outputs, could result in the increment of carbon emissions. While the tertiary industrial structure and the technology contribute to the suppression of carbon emissions in the YREB, however, their restrain effect is so weakening that cannot offset the growth of carbon emissions caused by their stimulating factors. Moreover, the result that urbanization is one of the contributors of promoting carbon emissions is consistent with previous studies [38]. Urbanization has a huge clustering effect on economic development, and there are intensive construction activities, extensive transportation systems and high population density in urban areas, which will increase energy consumption and carbon emissions. Thus the governments of the YREB ought to change its economic development model to intensive production and realize the de-coupling association between economic growth and carbon emissions [39].” 

In addition, we further perfect the minor editorial correction you mentioned.

Thank you very much for your careful review and precious comments to refine our manuscript once again. We hope the current version has resolved all the questions and look forward to your feedback. Please feel free to contact me at the address below if you have any questions.

Best regards,

Jingxue Zhang, Yanchao Feng*, Ziyi Zhu

Business School, Zhengzhou University, Zhengzhou 450001, PR China

* Correspondence: [email protected]; Tel.: +86-150-0218-2995

Round 2

Reviewer 3 Report

A significant improvement is clearly visible. The results are carefully explained and discussed which adds to the value of the paper.